# Do Long-Term Complications of Type 2 Diabetes Increase Susceptibility to Geriatric Syndromes in Older Adults?

**DOI:** 10.3390/medicina57090968

**Published:** 2021-09-15

**Authors:** Güzin Çakmak, Sencer Ganidağlı, Eyyüp Murat Efendioğlu, Ercüment Öztürk, Zeynel Abidin Öztürk

**Affiliations:** Department of Internal Medicine, Division of Geriatric Medicine, Faculty of Medicine, Gaziantep University, 27100 Gaziantep, Turkey; drsencer@yahoo.com (S.G.); eefendioglu@gmail.com (E.M.E.); ercument37@yahoo.com (E.Ö.); zaodr79@yahoo.com (Z.A.Ö.)

**Keywords:** diabetes mellitus, diabetes complications, frailty, sarcopenia, malnutrition, polypharmacy

## Abstract

*Background and Objectives:* Type 2 diabetes is one of the common chronic diseases in the elderly. It is thought that long-term complications of type 2 diabetes will negatively affect the quality of life in elderly individuals. It is possible that geriatric syndromes, especially frailty syndrome, are associated with diabetic complications, too. In this study, we aimed to evaluate the effect of macrovascular and microvascular complications of type 2 diabetes on frailty and other geriatric syndromes. In addition, the effect of these complications on quality of life was also reviewed. *Materials and Methods*: We conducted a cross-sectional study for four months. Comprehensive geriatric assessment tests were done on all patients. The Fried frailty index (FFI) was used for the evaluation of frailty syndrome. We assessed malnutrition by mini nutritional assessment short-form (MNA-SF), and Global Leadership Initiative on Malnutrition criteria (GLIM). The EWGSOP 2 criteria were used for the diagnosis of sarcopenia. Quality of life (QoL) was evaluated using the short form-36 (SF-36) questionnaire. Data analysis was done by SPSS version 22. *Results:* 237 females and 142 males with a mean age of 71.7 ± 6.1 years were included in the study. The frequency of macrovascular and microvascular complications was 41.4% and 68.1%, respectively. Frailty was found to be associated with macrovascular complications (*p* = 0.003). Handgrip strength, skeletal muscle mass index, and gait speed were decreased in the presence of macrovascular complications (*p* = 0.043, *p* < 0.001, *p* < 0.001). QoL was also decreased in patients with macrovascular complications (*p* = 0.003). Nutritional status and handgrip strength were negatively affected in patients with diabetic neuropathy (*p* = 0.019, *p* = 0.014). Polypharmacy was also found to be associated with macrovascular complications (*p* < 0.001, *p* < 0.001). Macrovascular complications were 2.5 times more common in malnourished patients according to GLIM and 3.2 times more common in patients with decreased gait speed. *Conclusion:* In this study, we observed that both macrovascular and microvascular complications of diabetes increase susceptibility to geriatric syndromes in elderly individuals. It could be useful to conduct prospective studies in which we can compare the effectiveness of treatment methods on this subject.

## 1. Introduction

Diabetes mellitus is a metabolic disorder characterized by chronic hyperglycemia caused by insufficient insulin secretion or insulin resistance [1]. The International Diabetes Federation has identified diabetes as one of the biggest health problems of the 21st century [2]. While the prevalence of diabetes patients worldwide was 8.5% in 2014, it is estimated that the number of people affected will increase from 422 million to 642 million by 2040 [3].

According to current data, there is an increase in both the number of elderly patients newly diagnosed with diabetes and the number of those who have been followed with diabetes for more than 20 years. Although type 2 diabetes is more common in elderly individuals, the presence of those diagnosed with type 1 diabetes and survived to 65 years old cannot be neglected. The follow-up of elderly diabetic individuals differs from those of young people. Especially the presence of comorbidities, geriatric syndromes, and the importance of hypoglycemia form the basis of this difference [4].

Diabetes has many complications. Acute metabolic complications such as diabetic ketoacidosis, hyperosmolar coma, and hypoglycemia are known to be associated with mortality. Microvascular and macrovascular complications caused by vascular damage due to hyperglycemia are also associated with mortality. Complications resulting from damage to small blood vessels due to prolonged hyperglycemia are defined as “microvascular disease” and complications caused by damage to the main arteries are defined as “macrovascular disease”. Diabetic retinopathy, nephropathy, and neuropathy are microvascular complications. Atherosclerotic diseases such as myocardial infarction and stroke are called macrovascular diseases [5].

Geriatric syndromes can be defined as common health problems caused by the breakdown of multiple systems in elderly people [6]. It is thought that diabetes mellitus accelerates the aging process and increases the incidence of geriatric syndromes due to the accumulation of advanced glycation end products. In addition, considering the total time spent hyperglycemic, we can say that diabetic complications are more common in elderly individuals.

Diabetes treatment in elderly individuals is more difficult due to physical and cognitive limitations. It is thought that the presence of geriatric syndromes makes it difficult to comply with the diet and treatment for diabetes [7]. Therefore, it is important to evaluate the relationship between geriatric syndromes and diabetes-related complications. There have been some studies in the past evaluating the relationship between diabetes and geriatric syndromes. For example, Yang et al., in their meta-analysis, revealed that the risk of falling is higher in elderly diabetic patients [8]. Mesinovic et al., on the other hand, demonstrated the relationship between diabetes and sarcopenia in elderly individuals [9]. The study of Vischer et al. can be cited as an example of studies evaluating the relationship between diabetes and malnutrition in elderly individuals [10]. Understandably, the relationship between diabetes and geriatric syndromes is also associated with diabetic complications. However, the number of studies that directly evaluate the relationship between geriatric syndromes and diabetic complications is very few.

Considering all this information, we aimed to evaluate the effect of macrovascular and microvascular complications of diabetes on geriatric syndromes such as frailty, sarcopenia, malnutrition, polypharmacy, and quality of life, in this study.

## 2. Materials and Methods

### 2.1. Participants

This cross-sectional study was carried out for a period of 4 months from August 2020 to December 2020. Diabetic patients admitted to the geriatric outpatient clinic were included in the study. The Local Research Ethics Committee approved the study. All participants gave informed consent.

### 2.2. Inclusion Criteria

Patients aged 65 and over and diagnosed with type 2 diabetes were included in this study. Those with newly diagnosed diabetes were not excluded from the study as complications may have developed during the period until the diagnosis.

### 2.3. Exclusion Criteria

Patients who had already been diagnosed with any of the geriatric syndromes such as sarcopenia, frailty, malnutrition were not included in the study. Patients who were in a cancer chemotherapy program or treated for an inflammatory disease were excluded too.

### 2.4. Comprehensive Geriatric Assessment

A cognitive evaluation was done by the standardized form of mini-mental state examination (MMSE), assessment of abilities of daily living (ADL) by Katz index, instrumental activities of daily living (IADL) by Lawton Brody index, risk of fall by Tinetti Balance-Gait Evaluation Scale and psychological status by the short form of Yesavage Geriatric Depression scale (GDS). All assessments were made by a skilled staff member.

In MMSE, patients were evaluated for six different areas; orientation, attention, registration, language, calculation, and recall [11]. Patients whose scores were ≤24 were accepted for the presence of dementia. Katz’s index of ADL appreciated patients for eating, personal hygiene, continence, dressing, feeding, and ambulating. Scores were between intervals of 0 and 6; high scores are considered high self-sufficiency [12]. Lawton Brody index was used for evaluating IADL like house cleaning, doing the laundry, marketing, managing medications, cooking, communicating with others, using transportation, and doing financial management; higher scores mean higher independence [13]. Tinetti Balance-Gait Evaluation Scale [14] and Timed up and Go Test (TUG) [15] were used for evaluating the falling risk. In the Tinetti Balance-Gait Evaluation Scale, a score of >24 means low risk of fall, 19–23 moderate risk of fall, and <19 high risks of fall [14]. Patients who took 14 s, or longer from the TUG test, were classified as high-risk for falling too [15]. GDS scores of 5 and higher are eligible for depression [16].

### 2.5. Nutritional Assessment

Malnutrition was assessed by a mini nutritional assessment, short-formed (MNA-SF) [17], and the Global Leadership Initiative on Malnutrition criteria (GLIM) [18]. MNA-SF is the first part of the mini nutritional assessment (MNA) test and takes only a few minutes to complete. The maximum score for MNA-SF is 14; a score of 12 points or greater indicates that the patient has an acceptable nutritional status and further assessment is not needed to be done. However, a score of 11 points or below is an indication to proceed with the GLIM. GLIM criteria are new criteria for diagnosing malnutrition. These criteria are based on the verification of phenotypic criteria, such as low body mass index, non-volitional loss of weight, or reduced muscle mass, associated with etiological variables, such as reduced food intake, the presence of inflammation, or disease burden. Also, the severity of malnutrition can be classified as moderate and severe according to body mass index, loss of weight, and muscle mass. 

### 2.6. Assessment for Sarcopenia

For defining sarcopenia, muscle strength, mass, and physical performances were assessed. SARC-F (strength, assistance walking, rising from a chair, climb stairs, and falls) test was used to select cases to evaluate muscle strength [19]. The handgrip test was performed if the patient had point ≥ 4 from SARC-F to diagnose probable sarcopenia. The handgrip test was performed by using a hand dynamometer with the dominant hand [20]. For females < 16 kg (kilograms), for males < 27 kg was accepted as probable sarcopenic. A bioimpedance test was carried out on probable sarcopenic patients to assess skeletal muscle mass. Sarcopenia was diagnosed by skeletal muscle mass index. In this study, we used skeletal muscle mass index (SMMI) adjusted to height. SMMI was calculated by dividing skeletal muscle mass by the square of height. SMMI cut-off values for sarcopenia were considered as 7.4 kg/m^2^ for women and 9.2 kg/m^2^ for men [21]. We evaluated gait speed with a four-meter gait speed test to diagnose severe sarcopenia. Patients with a gait speed below 0.8 m/s were defined as severe sarcopenic [22].

### 2.7. Assessment for Frailty Syndrome

Frailty was assessed by using the Fried Frailty Index (FFI). FFI is constituted from five criteria: unintentional weight loss (≥5% of body weight in the prior year), self-reported poor energy, weakness (grip strength in the lowest 20% at baseline), slow gait speed (the slowest 20% of the population was defined at baseline), and low physical activity (a weighted score of kilocalories expended per week was calculated at baseline, based on each patient’s report). People who were positive for three and more FFI criteria were defined as frail. Those who were positive for one or two criteria were described as pre-frail. People not meeting any criteria were confirmed defined as robust [23].

### 2.8. Assessment for Quality of Life

Quality of life (QoL) assessment was made with Short form-36 (SF 36). SF-36 is a quality-of-life assessment index developed by Rant Corporation in 1992. Quality of life was evaluated by 36 questions. Those questions were divided into eight domains including physical function capacity (PFC), physical role limitation (PRL), emotional role limitation (ERL), social function, pain, vitality, mental health, and general perception of health (PGH). Subscales evaluate QoL with a score between 0–100. The validity and reliability of the Turkish version of the test were made by Kocyiğit and colleagues [24].

### 2.9. Evaluation of Diabetes

Fasting plasma glucose (FPG) and HbA1c values were used for the diagnosis of diabetes. Blood samples were taken by venipuncture after overnight 12-h fasting. FPG was measured by the glucose oxidase method. HbA1c levels were measured in the same laboratory by high-performance liquid chromatography. Fasting plasma glucose levels of ≥126 and HbA1c of ≥6.5% were considered diabetic [25].

### 2.10. Evaluation of Microvascular Complications

A diagnosis of diabetic neuropathy was made by electroneuromyography (ENMG), and retinopathy by fundus examination. For the evaluation of diabetic neuropathy, primarily neurological examination was performed on the patients. Glove-sock hypoesthesia, decrease or loss of reflexes in the biceps, patella, or Achilles, and motor loss in at least two extremities were considered as clinical polyneuropathy.

Subnervous conduction velocity (NCV) was evaluated by electroneuromyography as motor conduction velocity (MCV) and sensory conduction velocity (SCV). The NCV limit for the upper extremity was accepted as 50 m/s for MCV and 43 m/s for SCV. In addition, the NCV limit for the lower extremity was 42 m/s for MCV and SCV. The limit for motor unit potential (MUP) amplitude was accepted as 6 mV for the median and ulnar nerve, 3 mV for the peroneal nerve, and 4 mV for the tibial nerve. Sensory nerve action potential amplitude was taken as 10 μV for median and ulnar nerves and 6 μV for sural and peroneal superficial nerves. The upper limit of the tibial F wave was accepted as 55 ms. Polyneuropathy was divided into three classes as motor, sensory and sensorimotor according to the involvement of sensory and motor nerves. In terms of pathogenesis, it was divided into groups as demyelination polyneuropathy, axonal polyneuropathy, or mixed polyneuropathy. Demyelination polyneuropathy was defined as more than 30% prolongation of motor distal latency, more than 25% decrease in conduction velocity, more than 55 ms prolongation of the F wave, or temporal dispersion (proximal/distal MUP duration > 1.15). The presence of motor and sensory amplitude reduction and/or denervation potentials greater than 40% of the normal value in the ENMG needle was evaluated as axonal polyneuropathy [26].

We determined the presence of nephropathy by the spot urine albumin/creatinine ratio is positive (>30 mg albumin per gram of creatinine) in two out of three tests over six months. Urine albumin and creatinine levels were measured by turbidimetric immunoassay and photometric assays respectively [27].

### 2.11. Statistical Analysis

The variables were analyzed for their distribution normality using the Kolmogorov–Smirnov and Shapiro–Wilk test. All data were disturbed normally (*p* > 0.05). Power analysis was done by Gpower 3.9.1 software. To find statistically significant, the expectation that a medium effect size (dz = 0.5) will occur between the parameters, the minimum number required was determined as 80 (α = 0.05; 1 − β = 0.80). Descriptive statistics are given for continuous variables. Continuous variables of groups were assessed by using the independent sample *t*-test and analysis of variance (ANOVA). The data were expressed as mean ± deviation standard deviation (S.D.). Relationships between parameters were investigated by the chi-square test and Pearson correlation analysis. We used multinominal logistic regression to simulate a model to determine factors affecting the presence of macrovascular complications. The statistical significance level was determined as *p* < 0.05. We used SPSS version 22.0 (IBM, Armonk, NY, USA) to analyze the data.

## 3. Results

The study population was composed of 237 women and 142 men, of mean age 71.7 +/− 6.1 years. Age and gender were not related to the presence of macrovascular and microvascular complications (*p* > 0.05). There was no significant relationship between education, place of residence, marital status, and the presence of microvascular or macrovascular complications (*p* > 0.5). Smoking was associated with the presence of macrovascular complications (*p* < 0.001). While the rate of smokers in those who suffered from macrovascular complications was 16%, it was 12% in those who did not. Demographic data about the patients are summarized in Table 1. Macrovascular complications were present in 41.4% (*n* = 157), microvascular complications were present in 68.1% (*n* = 258) of them. Fourteen-point- five percent of the patients had only peripheral artery disease, 0.8% had only cerebrovascular disease, 9.8% had only coronary artery disease, 0.8% had peripheral artery disease and cerebrovascular disease, 1.3% had coronary artery disease and cerebrovascular disease, 12.9% had peripheral artery disease and coronary artery disease. One point three percent of patients had all three. It was observed that ADL and IADL were preserved better in patients without macrovascular complications (*p* < 0.001, *p* = 0.01). Nutritional status evaluated by MNA-SF and GLIM was also better in the group without macrovascular complications (*p* < 0.001, *p* = 0.014). Frailty syndrome was associated with macrovascular complications (*p* = 0.003). It was observed that the number of drugs used was higher in patients with macrovascular complications (*p* < 0.001). Polypharmacy was seen in 207 (55%) of the patients. Macrovascular complications were seen in 53.6% of those with polypharmacy, and 26.2% of those without polypharmacy (r = 0.280, *p* < 0.001). Microvascular complications were seen in 70% of those with polypharmacy and 66.7% of those without polypharmacy (*p* = 0.838). QoL was also decreased in patients with macrovascular complications (*p* = 0.003). Macrovascular complications were not related to depression, dementia, or balance. Results were summarized in Table 2. While the calf circumference is higher in those without macrovascular complications, the upper-middle arm circumference is higher in those with macrovascular complications (*p* = 0.001, *p* = 0.022). Handgrip strength, skeletal muscle mass index, and gait speed were lower in those with macrovascular complications (*p* = 0.043, *p* < 0.001, *p* < 0.001). Results were depicted in Table 3. The relationship between subgroups of macrovascular complications and geriatric syndromes is summarized in Table 4. Nutritional status evaluated by GLIM and handgrip strength were worse in patients with diabetic neuropathy than others (*p* = 0.019, *p* = 0.014). The mean handgrip strength was calculated as 16.3 ± 5.9 kg in those with diabetic neuropathy, and 18.4 ± 6.4 kg in those without. According to GLIM, the malnutrition rate is 56% in those with neuropathy and 43% in those who do not. There was no significant relationship between disability, cognitive functions, depression and balance, and microvascular complications (*p* > 0.05). We found that GLIM score, SMMI, gait speed, and the number of drugs used were independently related to macrovascular complications in multinominal logistic regression analysis (R^2^ = 0.296, *p* < 0.001). We found that macrovascular complications were 2.5 times more common in patients with malnutrition according to GLIM and 3.2 times more common in patients with decreased gait speed. Results of logistic regression analysis were summarized in Table 5.

HbA1c values were 6.5–7.9% in 28.8% of the patients, 8–9.9% in 51.7%, and >10% in 19.5% of the patients. The mean HbA1c value was 8.6%. Blood glucose regulation was found to be associated with ADL, IADL, sarcopenia, and malnutrition. In the group with HbA1c above 10%, ADL, IADL, and nutritional status have been impaired. Sarcopenia was seen to be increased with increasing HbA1c values. The results are summarized in Table 6. Drugs used in the treatment of diabetes were found to be related to ADL and sarcopenia. In our study, sarcopenia and impaired daily living activities were seen at least in the group using metformin, and then in those using other oral antidiabetics in addition to metformin or only. The results are summarized in Table 7.

## 4. Discussion

According to the results of our study, the macrovascular complications of diabetes were found to be associated with disability, malnutrition, sarcopenia, polypharmacy, and decreased quality of life. Among microvascular complications, neuropathy is associated with possible sarcopenia and malnutrition.

In diabetic patients, the risk of acute myocardial infarction has increased 2.13 times in men and 2.95 times in women. The risk of cardiovascular disease in diabetic individuals has increased 2–4 times compared to the general population. Diabetes is also a risk factor for ischemic stroke [28].

Diabetic macroangiopathy causes damage to the cardiovascular, central, and peripheral nervous systems and the vessels of the extremities. Because of that situation incidence of cardiovascular disease is among the leading causes of death in patients with diabetes. Atherosclerotic conditions such as cerebrovascular disease, peripheral artery disease, and sequelae related to these are also common [29]. It has also been shown in previous studies that there may be a decrease in cognition and functional capacity in macrovascular disease [30].

The prevalence of peripheral artery disease in diabetic patients is 20%. Approximately one-third of peripheral arterial diseases present with intermittent claudication. This is likely to affect gait speed as well. Dyslipidemia is seen in approximately 70–80% of diabetic patients. Insulin resistance increases free fatty acid production and triglyceride synthesis in the liver. This results in hepatic steatosis. In the advanced stage, hepatic steatosis turns into steatohepatitis due to the toxic properties of fats. Excess lipids degraded in mitochondria increase oxidative stress and inflammation. Oxidative stress and inflammation form the basis of the pathogenesis of all macrovascular complications [28].

Co-occurrence of cardiovascular disease, which is one of the diabetic macrovascular complications, and nephropathy, which is one of the microvascular complications, is quite common. Impaired blood pressure regulation with the development of diabetic nephropathy, increased proteinuria, inflammation, and oxidative stress are the main causes of this. In our study, however, the relationship between cardiovascular diseases and geriatric syndromes could not be seen with nephropathy [31].

In another study, it was shown that diabetic macrovascular complications increase the risk of delirium and falls [32]. In contrast to this, we did not find a relationship between balance tests and the presence of macrovascular complications in our study.

Oral candida, gastroparesis, nonalcoholic fatty liver disease, gastroesophageal reflux, dysphagia, and chronic diarrhea caused by diabetes have the potential to lead to nutritional disorders. Early satiety caused by gastroparesis, fullness after meals, nausea, vomiting with undigested food, bloating and abdominal pain cause both a decrease in appetite and ineffective use of the consumed food [33]. Colorectal dysfunctions are common in patients with a history of uncontrolled diabetes. Constipation disrupts the quality of life and eating habits, especially in patients with diabetic enteropathy [34]. In addition to all these, exocrine pancreatic insufficiency, especially seen in uncontrolled type 2 diabetes, contributes to nutritional disorders related to diabetes [35]. In the light of this information, it can be predicted that diabetic microvascular damage, especially diabetic neuropathy, may lead to nutritional disorders. In our study, both macrovascular complications and the microvascular complication, diabetic neuropathy, were found to be associated with malnutrition.

In our study, sarcopenia was associated with macrovascular damage, and probable sarcopenia was related to microvascular damage. Sarcopenia seen in diabetes has primarily been associated with malnutrition [36]. Muscle wasting in diabetes has also been associated with inflammation and oxidative stress [37].

As the criteria of the FFI index such as weakness, slowness, low physical activity level is common with sarcopenia, diabetes, which is known to affect sarcopenia, is expected to affect frailty [38]. However, this effect could not be demonstrated in our study.

Richardson et al. reported that multimorbidity, which increases in frequency with the presence of diabetes, results in polypharmacy and increases the use of psychotropic drugs [39]. In our study, macrovascular damage was found to be related to polypharmacy. This relation is thought to be associated with comorbidities such as atherosclerotic heart disease and cerebrovascular disease caused by macrovascular damage. We do not consider associating this situation with psychotropic drugs because we cannot find a relationship with GDS.

In this study, we concluded that quality of life is associated with macrovascular complications of diabetes. The relationship of both macrovascular and microvascular complications with quality of life has been mentioned before. End organ damages, amputations, and intensive treatment modalities are some of the reasons for this effect [40]. Stojanovic et al. demonstrated a similar relationship between the diabetes complications such as angina pectoris, heart failure, diabetes nephropathy, diabetes retinopathy, and the quality of life both by SF-36 and the EQ-VAS score [41].

The lower incidence of geriatric syndromes in those taking metformin and other oral antidiabetic agents was attributed to the better blood glucose regulation of the patients in these groups. Data on the relationship between geriatric syndromes and HbA1c values confirm this.

The most important approach to prevent geriatric syndromes in diabetic patients is to provide blood glucose regulation. In older adults, blood glucose regulation should be ensured in the light of current guidelines, avoiding inappropriate drug use and considering ease of drug use. In addition, precautions taken for elderly individuals for frailty, disability, sarcopenia, and malnutrition should be followed more closely in diabetic patients. Attention should be paid to diet and exercise, and insufficient protein intake should be avoided while dieting. Again, the use of antioxidants and anti-inflammatories can be considered to prevent oxidative stress and inflammation [42].

The strength of our study is that there is no study in the literature evaluating the relationship between nearly all geriatric syndromes and macrovascular and microvascular diabetes complications. There are review articles that have evaluated the results of previous studies conducted to evaluate the relationship of separate geriatric syndromes with different diabetic complications.

In our study, the relationship of geriatric syndromes with macrovascular complications has been revealed more clearly than microvascular complications. Although the reason for this is thought to be the occurrence of macrovascular complications later than microvascular complications, the lack of age difference between the two groups creates a conflict in this regard. We decided to explain this with the presence of a small number of patients without microvascular complications. More significant results could have been obtained, especially if microvascular complications were studied in a larger population with a more balanced control group. This is the most important limitation of our study.

## 5. Conclusions

Both diabetes and complications related to diabetes are more common in the geriatric age group. Studies have concluded that these complications increase the frequency of geriatric syndromes. Good blood glucose regulation and prevention of complications in diabetic patients will decrease the frequency of geriatric syndromes. Prospective studies that evaluate the effect of good diabetes management on geriatric syndromes will also be useful.

## Figures and Tables

**Table 1 medicina-57-00968-t001:** Demographic features of the groups.

Parameters	Macrovascular Complications Present (*n*= 157)	Microvascular Complications Present (*n* = 258)	Macrovascular or Microvascular Complications Absent (*n* = 77)	*p*-Value
Age (mean ± SD)	71.5 ± 5.3	71.4 ± 6.3	72.8 ± 6.3	0.187
Gender (%)				
Female	64.3	64	61	
Male	35.7	36	39	0.831
Education (%)				
Illiterate	51	53	39.4	
Primary school graduate	33.8	33	39.4	
Secondary school graduate	5.1	5	9.6	
High school graduate	7	6	6.6	
Graduated from university	3.2	3	5	0.378
Place of residence (%)				
Lives alone	14	14	6.5	
Lives with spouse	72	74	45.5	
Lives with relatives	14	11	48	
Lives with caregiver	0	1	0	
Marital status (%)				
Single	28	26	14.3	
Married	72	74	85.7	0.925
Smoking (%)				
Yes	15.9	19	12.9	
No	84.1	81	87.1	0.306

**Table 2 medicina-57-00968-t002:** Results of comprehensive geriatric assessment, frailty, and QOL tests.

Tests	Macrovascular Complications Present (*n* = 157)	Macrovascular Complications Absent (*n* = 222)	*p*-Value
ADL (mean ± SD)	4.2 ± 1.4	4.9 ± 1.2	<0.001
IADL (mean ± SD)	4.7 ± 1.7	5.2 ± 1.9	0.01
sMMT (mean ± SD)	24.8 ± 4.5	25.3 ± 4.0	0.341
GDS (mean ± SD)	5.24 ± 4.3	4.94 ± 4.7	0.535
MNA-SF (mean ± SD)	12.0 ± 2.2	11.1 ± 2.6	<0.001
GLIM (mean ± SD)	0.47 ± 0.5	0.62 ± 0.6	0.014
Tinetti (mean ± SD)	22.1 ± 5.4	21.2 ± 6.0	0.164
TUG (s) (mean ± SD)	12.4 ± 5.9	11.4 ± 5.8	0.167
FFI (mean ± SD)	1.8 ± 1.1	1.5 ± 1.2	0.003
SF-36 (mean ± SD)	374 ± 142	428 ± 146	0.003
Number of drugs used (mean ± SD)	5.0 ± 2.8	3.5 ± 2.5	<0.001

ADL: activities of daily living, IADL: instrumental activities of daily living, sMMT: standardized mini-mental test, GDS: geriatric depression scale, MNA-SF: mini-nutritional assessment short form, GLIM: global leadership initiative on malnutrition, TUG: timed up and go test, FFI: Fried frailty index, s: second.

**Table 3 medicina-57-00968-t003:** Results of sarcopenia assessments.

Measurements	Macrovascular Complications Present (*n* = 157)	Macrovascular Complications Absent (*n* = 222)	*p*-Value
mid-arm circumference (cm)	31.7 ± 4.1	30.7 ± 4.6	0.022
calf circumference (cm)	38.7 ± 6.5	42.4 ± 12.5	0.001
handgrip strength (kg)	20.5 ± 8.5	22.8 ± 10.1	0.027
SMM (kg)	29.2 ± 9.3	36.8 ± 12	<0.001
SMMI (kg/m^2^)	9.8 ± 3.0	12.1 ± 3.8	<0.001
gait speed (m/s)	0.8 ± 0.4	1.2 ± 0.8	<0.001

cm: centimeter, kg: kilogram, kg/m^2^: kilogram per square-meter, m/s: meter per second, SMM: skeletal muscle mass, SMMI: skeletal muscle mass index.

**Table 4 medicina-57-00968-t004:** Relationship between subgroups of macrovascular complications and geriatric syndromes.

	PAD (*n* = 55)	CAD (*n* = 3)	CVD (*n* = 37)	More than one Macrovascular Complication (*n* = 62)	Macrovascular Complications Absent (*n* = 222)	*p*-Value
ADL (mean ± SD)	4.1 ± 1.4	4.5 ± 1.1	3.8 ± 1.6	3.7 ± 1.5	4.9 ± 1.2	<0.001
IADL(mean ± SD)	5.0 ± 1.6	4.5 ± 1.5	4.3 ± 1.2	4.6 ± 1.8	5.2 ± 1.9	0.04
calf circumference (cm) (mean ± SD)	39.2 ± 6.5	37.1 ± 4.9	36.3 ± 2.5	39.4 ± 7.5	42.4 ± 12.5	0.014
handgrip strength (kg) (mean ± SD)	18.9 ± 8.5	22.9 ± 9.1	22.1 ± 11.8	20.4 ± 7.9	22.8 ± 10.1	<0.001
SMMI (kg/m^2^) (mean ± SD)	9.7 ± 2.9	10.2 ± 3.5	11 ± 2.3	9.5 ± 3	12.1 ± 3.8	<0.001
gait speed (m/s) (mean ± SD)	0.8 ± 0.4	0.8 ± 0.4	0.6 ± 0.2	0.8 ± 0.4	1 ± 0.8	<0.001
MNA-SF (mean ± SD)	12 ± 2.2	12.2 ± 1.8	13.7 ± 0.6	11.8 ± 2.4	11.1 ± 2.6	0.008
GLIM (mean ± SD)	0.4 ± 0.5	0.6 ± 0.6	0.0 ± 0.0	0.5 ± 0.5	0.6 ± 0.6	0.026
FFI (mean ± SD)	2.0 ± 1.2	1.5 ± 1.1	1.3 ± 0.6	2 ± 1.2	1.5 ± 1.2	0.008
Number of drugs used (mean ± SD)	3.8 ± 2	4.7 ± 2.8	5.3 ± 2.5	6.1 ± 3.1	3.5 ± 2.5	<0.001
SF-36 (mean ± SD)	394 ± 144	377.1 ± 117.2	396.6 ± 92.8	350.5 ± 157.7	429.1 ±146.4	0.028

cm: centimeter, kg: kilogram, kg/m^2^: kilogram per square -meter, m/s: meter per second, ADL: activities of daily living, IADL: instrumental activities of daily living, SMM: skeletal muscle mass, SMMI: skeletal muscle mass index, GLIM: global leadership initiative on malnutrition, MNA-SF: mini-nutritional assessment short form, FFI: Fried frailty index, PAD: Peripheral artery disease, CAD: Coronary artery disease, CVD: Cerebrovascular disease.

**Table 5 medicina-57-00968-t005:** Results of logistic regression analysis.

GLIM	0.937	12.893	2.55	<0.001
SMMI	0.220	21.539	1.24	<0.001
FFI	0.023	0.025	1.023	0.874
Hand grip strength (kg)	−0.012	0.523	0.988	0.470
Gait speed (m/s)	1.183	11.086	3.264	0.001
Number of drugs used	−0.201	17.446	0.818	<0.001

kg: kilogram, m/s: meter per second, GLIM: global leadership initiative on malnutrition, FFI: Fried frailty index.

**Table 6 medicina-57-00968-t006:** Relationship between blood glucose regulation and geriatric syndromes.

HbA1c Value	6.5–7.9% (*n* = 109) (28.8%)	8–9.9% (*n* = 196) (51.7%)	>10% (*n* = 74)	*p*-Value
ADL (mean ± SD)	5.9 ± 1.3	6.3 ± 0.9	5.5 ± 1.6	0.002
IADL (mean ± SD)	5.1 ± 1.8	5.2 ± 1.9	4.5 ± 1.7	0.002
handgrip strength (kg) (mean ± SD)	25.6 ± 9.7	21.2 ± 9.3	18 ± 7.8	<0.001
SMMI (kg/m^2^) (mean ± SD)	12 ± 3.4	11.1 ± 3.9	10 ± 3.8	0.002
gait speed (m/s) (mean ± SD)	1.2 ± 0.9	1 ± 0.7	0.8 ± 0.5	<0.001
MNA-SF (mean ± SD)	12.3 ± 1.9	11.4 ± 2.4	11.3 ± 2.5	0.01

cm: centimeter, kg: kilogram, kg/m^2^: kilogram per square meter, m/s: meter per second, ADL: activities of daily living, IADL: instrumental activities of daily living, SMM: skeletal muscle mass, SMMI: skeletal muscle mass index, MNA-SF: mini-nutritional assessment short form.

**Table 7 medicina-57-00968-t007:** Relationship between antidiabetic drugs and geriatric syndromes.

Drugs Used for Diabetes	Metformin	Metformin and/or Other Oral Antidiabetics	Oral Antidiabetics+Insulin	Insulin	*p*-Value
ADL (mean ± SD)	6.4 ± 1.9	6.1 ± 1.7	5.5 ± 1.7	5.7 ± 1.8	<0.001
handgrip strength (kg) (mean ± SD)	25.1 ± 11.8	23.5 ± 9.5	18.1 ± 6.1	18.1 ± 8.7	<0.001
SMMI (kg/m^2^) (mean ± SD)	12.1 ± 2.8	11.7 ± 3. 8	10.1 ± 3.5	10 ± 3.7	<0.001
gait speed (m/s) (mean ± SD)	1.5 ± 1.1	1 ± 0.6	0.9 ± 0.6	0.9 ± 0.6	<0.001
calf circumfence (cm) (mean ± SD)	43.9 ± 16	39.2 ± 9	42.7 ± 11.2	41.4 ± 8.8	0.01

cm: centimeter, kg: kilogram, kg/m^2^: kilogram per square meter, m/s: meter per second, ADL: activities of daily living, SMM: skeletal muscle mass, SMMI: skeletal muscle mass index.

## Data Availability

The study did not report any data.

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
