# Peer review of "Do Long-Term Complications of Type 2 Diabetes Increase Susceptibility to Geriatric Syndromes in Older Adults?"

_medicina, 2021, doi:10.3390/medicina57090968_

Round 1

Reviewer 1 Report

This study aimed to evaluate the effect of macrovascular and microvascular complications of diabetes on geriatric syndromes such as frailty, sarcope- nia, malnutrition, polypharmacy, and quality of life. The paper is very poorly written.

Some of the serious  improvements that must be done: 

Shape requests

Extensively revision of the shape of the paper according to the Instructions for authors - check and proceed accordingly https://www.mdpi.com/journal/medicina/instructions The Instructions for authors must be carefully read and understand. This is why they are provided by the journal, to be followed!

Please move the paper on the draft recommended by Medicina MDPI https://www.mdpi.com/journal/medicina/instructions Accepted File Formats Authors must use the Microsoft Word template....

Number the sections / subsections and set all the text according to the same Instructions for authors. Respect the size of the characters.

Respect the format of the Tables requested by the journal. In the head of the tables not p= but p values

Acronyms/Abbreviations/Initialisms should be defined the first time they appear in each of three sections: the abstract; the main text; the first figure or table. When defined for the first time, the acronym/abbreviation/initialism should be added in parentheses after the written-out form. Proceed accordingly in the manuscript and under each table.

Content requests

As the last paragraph of Introduction section please highlight better the novel or special aspects that this paper brings to the field, a lot of papers being already published regarding this topic. You must draw attention on your paper of those interested by this topic. Why you chose it? what is the difference between other papers and this one?

Results. Very poor. More research must be done. Did the geriatric complications in your study correlate with glycemic control? Please detail:

  • the statistics on each diabetic type of macro vascular complications: acute coronary syndrome, stroke, peripheral artery disease;
  • the groups of patients regarding their anti-diabetic therapy glucose control, microvascular and macro vascular complications.

The Discussion chapter should be improved, as it is very poor as well. Please discuss further:

  • the pathophysiologic mechanisms regarding the apparition of geriatric complications in DM. Please check and refer to Vesa et al. Current Data Regarding the Relationship between Type 2 Diabetes Mellitus and Cardiovascular Risk Factors. Diagnostics 2020, 10, 314.  https://doi.org/10.3390/diagnostics10050314
  • the impact of certain anti-diabetic medications increase or decrease of risk of apparition of such geriatric complications;
  • the correlation between geriatric complications of DM and cardiovascular risk. l suggest Moisi et al. Acute Coronary Syndromes in Chronic Kidney Disease: Clinical and Therapeutic Characteristics. Medicina 2020, 56, 118. https://doi.org/10.3390/medicina56030118 and Babes et al. Value of Hematological and Coagulation Parameters as Prognostic Factors in Acute Coronary Syndromes. Diagnostics 2021, 11, 850. https://doi.org/10.3390/diagnostics11050850
  • certain modalities for prevention of geriatric complications, sarcopenia, muscular force increase and increase of autonomy for diabetic patients

Author Response

Dear editor,

We appreciate the time and effort you have dedicated to providing insightful feedback on ways to strengthen our paper. 

Based on your suggestions, I edited the article to "accepted file format". 

Although there are similar studies that have been done before, I have summarized the reasons why I have done this study in the introduction part.

In addition to these, I made some additions to my analysis. I also examined the relationship between cardiovascular disease, cerebrovascular disease, peripheral artery disease and their combinations and geriatric syndromes and summarized the results in table 4. I analyzed the relationship between blood glucose regulation status and geriatric syndromes and summarized the results in table 6. I analyzed the relationship between treatment used for diabetes and geriatric syndromes and summarized the results in table 7. I also made additions to the discussion section according to your suggestions.

Best regards,

Reviewer 2 Report

Cakma et al. evaluated the effects of macrovascular and microvascular complications of type 2 diabetes on frailty and other geriatric syndromes. The manuscript is comprehensible with no misleading information.

However, the sample size is relatively small to assess these effects. Moreover, even if it did, it has already been well-established that macrovascular and microvascular complications affect frailty and other geriatric syndromes. Hence, I see no novel information arising from this manuscript that would contribute to scientific community in this field.

I would suggest the authors to represent some of the relevant data in figures and, more importantly, to add some novel perspective which might make this manuscript suitable for publication.

Author Response

Although there are similar studies that have been done before, I summarized the reasons why I have done this study in the introduction part.

In addition to these, I made some additions to my analysis. I also examined the relationship between cardiovascular disease, cerebrovascular disease, peripheral artery disease and their combinations and geriatric syndromes and summarized the results in table 4. I analyzed the relationship between blood glucose regulation status and geriatric syndromes and summarized the results in table 6. I analyzed the relationship between treatment used for diabetes and geriatric syndromes and summarized the results in table 7. I also made additions to the discussion section.

Round 2

Reviewer 1 Report

All the modifications have been done. I reccomend publication. Comgratulations!

Reviewer 2 Report

No further comments.